# ER²Score: An Explainable and Customizable Metric for Assessing Radiology Reports with LLM-based Rewards

## Abstract

Recently, the automated generation of radiology reports (R2Gen) has seen considerable growth, introducing new challenges in evaluation due to its complex nature. Traditional metrics often fail to provide accurate evaluations due to their reliance on rigid word-matching techniques or their exclusive focus on pathological entities, leading to inconsistencies with human assessments. To bridge this gap, we introduce ER²Score, an automatic evaluation metric designed specifically for R2Gen that harnesses the capabilities of Large Language Models (LLMs). Our metric leverages a reward model and a tailored design for training data, allowing customization of evaluation criteria based on user-defined needs. It not only scores reports according to user-specified criteria but also provides detailed sub-scores, enhancing interpretability and allowing users to adjust the criteria between different aspects of reports. Leveraging GPT-4, we generate extensive evaluation data for training based on two different scoring systems, respectively, including reports of varying quality alongside corresponding scores. These GPT-generated reports are then paired as accepted and rejected samples to train an LLM towards a reward model, which assigns higher rewards to the report with high quality. Our proposed loss function enables this model to simultaneously output multiple individual rewards corresponding to the number of evaluation criteria, with their summation as our final ER²Score. Our experiments demonstrate ER²Score's heightened correlation with human judgments and superior performance in model selection compared to traditional metrics. Notably, our model's capability to provide not only a single overall score but also scores for individual evaluation items enhances the interpretability of the assessment results. We also showcase the flexible training of our model to varying evaluation systems. We will release the code on GitHub.

## 1 Introduction

In recent years, automated radiology report generation (R2Gen) has experienced significant expansion. This intricate AI task demands a profound comprehension of clinically relevant high-level semantics, presenting challenges not only in the generation process but also in evaluating the quality of the output reports. Automated assessment of radiology report generation typically involves metrics gauging the semantic accuracy of the generated reports against the reference reports. Traditional natural language generation (NLG) metrics, such as the BLEU metric (Papineni et al., 2002) and METEOR (Reimers & Gurevych, 2019), primarily quantify n-gram matches, often overlooking important factors like lexical and structural diversity, which are essential for capturing the true meaning of the reports. These n-gram-based evaluation metrics are often criticized as misjudging paraphrasing and failing to capture complex diagnostic information adequately. To address these issues, approaches like BERTScore (Zhang et al., 2020) have been proposed, utilizing contextualized token embedding to detect paraphrasing more effectively. Furthermore, comprehensive evaluations now often incorporate clinically relevant scores, such as F1 scores of pathological entities labeled by CheXbert (Smit et al., 2020) or Radgraph (Jain et al., 2021). However, these clinical scores are constrained by their predefined set of pathological entities and encounter challenges in accurately assessing the correlations among these entities. Despite efforts to improve the evaluation of report generation, existing evaluation metrics often do not align well with human judgment (Liu et al.,

2024a). A recent work Yu et al. (2023a) proposed the RadCliQ score, which linearly combines multiple existing metrics while regressing combination weights from human-marked error scores to better align with human evaluation. However, RadCliQ's reliance on a limited set of expensive human-annotated training samples poses a challenge. On the other hand, while recent advances in Large Language Models (LLMs), like GPT-4 (OpenAI, 2023), suggest their potential for report evaluation with proper prompts, direct applying GPT-4 for this purpose may be impractical. It raises privacy concerns due to the need for online evaluation and demands substantial computing resources, considering its size and general-purpose nature, which may not be cost-effective for R2Gen.

To drive progress in this field, this study proposes $\text{ER}^2\text{Score}$, an innovative metric tailored specifically for evaluating automated radiology report generation. Leveraging GPT-4's human-like scoring capacity (Chiang & Lee, 2023; Liu et al., 2024b), our method autonomously produces evaluation samples that mimic human judgment. These samples are subsequently utilized to train an LLM-based reward model for automated scoring. In comparison to traditional evaluation metrics, $\text{ER}^2\text{Score}$ substantially improves the alignment with human assessments, leading to a more precise evaluation of report quality. Moreover, instead of merely providing an overall score, our model simultaneously outputs the scores for individual evaluation criteria, improving the interpretability of the assessment results. For example, by combining sub-criteria, we can clearly identify the reasons for a report's poor quality, e.g., whether due to incorrect lesion location, incorrect severity of findings, or omission of findings. Meanwhile, by generating training samples using LLMs, our method reduces the dependence on costly human annotations, enabling scalable model training and greater flexibility in adapting to different evaluation criteria. To operationalize our approach, we utilized two distinct sets of evaluation criteria (scoring systems) in this study. Utilizing the defined criteria, we prompt GPT-4 to generate report samples with varied quality levels, pairing reports of different quality corresponding to the same ground-truth report as "accepted" and "rejected" samples with score margins. These paired samples were then used to fine-tune the pretrained Llama3 model (Meta, 2024)using reward modeling techniques. Our proposed loss function enables this model to produce multiple individual rewards concurrently, each corresponding to one evaluation criteria, which are then summed to produce our final $\text{ER}^2\text{Score}$. Validating our model on two datasets paired with human evaluations, we found $\text{ER}^2\text{Score}$ aligns more closely with human judgment than other traditional metrics and exhibits versatility to accommodate different evaluation criteria.

Our main contributions are summarized as follows:

(1) Our study presents a novel approach to training LLMs to generate $\text{ER}^2\text{Score}$, a human-consistent metric designed for automated radiology report evaluation. Through our novel loss function discerning report rankings, we finetune LLMs to produce rewards aligned with our scoring system in a fine-grained manner, enhancing alignment with human evaluations and bolstering assessment accuracy.

(2) Importantly, our evaluation metrics assesses not only the overall score for a report but also concurrently the detailed sub-scores based on diverse criteria. This capability, to the best of our knowledge, has not been achieved by existing evaluation metrics. It enhances the interpretability of the evaluation, enabling users to discern specific aspects influencing the overall score.

(3) By facilitating a tailored analysis of report components, our $\text{ER}^2\text{Score}$ allows users to customize the evaluation framework to suit their specific needs. This level of customization could contribute to more targeted improvements in report generation. This capacity of $\text{ER}^2\text{Score}$ has been demonstrated by its versatility to accommodates two distinct sets of evaluation standards, respectively.

## 2 METHOD

Traditional NLP evaluation metrics typically assess the similarity between a machine-generated report $x$ and a reference report $\hat{x}$ using n-gram overlap. However, these metrics often fail to capture the semantic equivalence and clinical relevance essential for accurate radiology report evaluation. To address these shortcomings, we introduce a new evaluation metric that better reflects the semantic content and clinical significance of the reports, aligning closely with human assessments.

Our model not only provides an overall score but also delivers nuanced sub-scores to facilitate a more detailed interpretation of the assessment. This approach leverages GPT-4 to generate training samples by scoring $x$ against its reference $\hat{x}$ based on specified criteria. These samples are then used to train a reward model with our proposed reward loss function to predict sub-scores. The summation of these sub-scores results in the final overall score. The overview of our framework is presented in Figure 1.

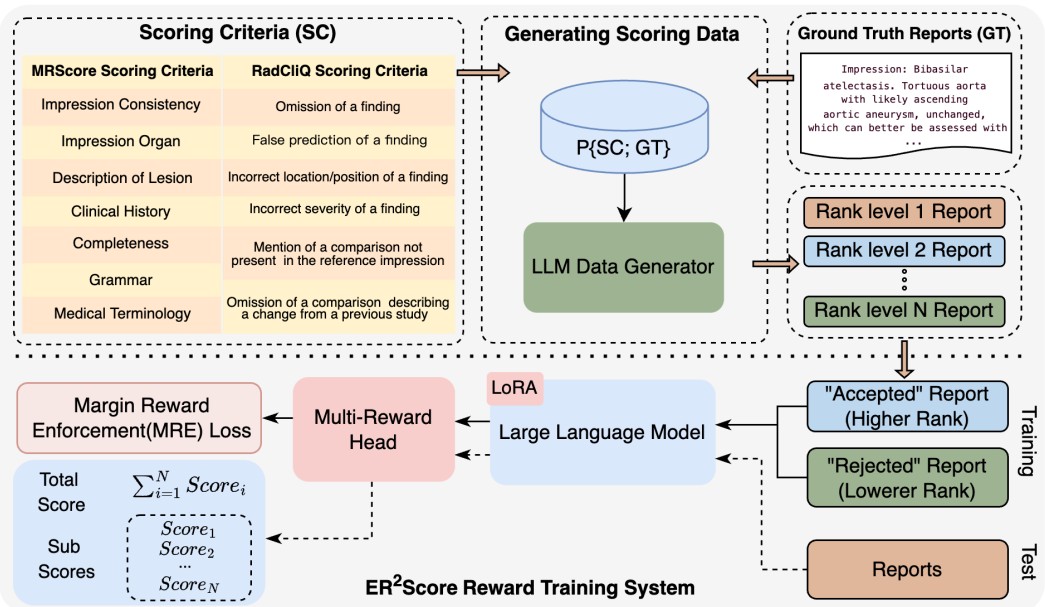

Figure 1: Overview of ER²Score. The upper portion illustrates the training data generation process, while the lower portion represents the training process for the reward model using LoRA. In the lower portion, the solid line indicates the training phase while the dashed line indicates the inference phase.

## 2.1 GENERATING TRAINING DATA BY GPT-4

Recent studies have demonstrated GPT-4's capability in evaluating chest X-ray reports. When prompted with specified criteria, **GPT-4 can generate similarity assessments that statistically correlate with human evaluations**, as consistently verified in Chiang & Lee (2023) and Liu et al. (2024b). For example, in Chiang & Lee (2023), GPT-4 achieved Kendall's tau of 0.735 with radiologists' annotations using RadCliQ scoring system. In Liu et al. (2024b), GPT-4 scored a Kendall-Tau correlation of 0.531 with human ratings using MRScore scoring system. Building on this observation, we utilize GPT-4 to generate extensive scoring data, including both reports and the corresponding scores, for training purposes. The process is elaborated as follows.

**Defining Scoring Criteria** Various assessment criteria have been reported in the literature. In this study, we investigate two scoring systems to demonstrate our model's versatility across different evaluation rules. The RadCliQ scoring system proposed in Yu et al. (2023a) evaluates both clinically significant and insignificant errors across six error categories: 1) false prediction of a finding, 2) omission of a finding, 3) incorrect location or position of a finding, 4) incorrect severity of a finding, 5) mention of a comparison absent in the reference impression, and 6) omission of a comparison that notes a change from a previous study. The total score is the sum of the error counts, highlighting the importance of clinical findings. Differently, the MRScore scoring system proposed in Liu et al. (2024b) addresses both clinical findings and linguistic concerns. It involves seven fundamental items from Radiologists' expertise and literature review: "impression consistency", "impression organs", "description of lesions," "clinical history", "completeness", "grammar", and "medical terminology", with a detailed explanation. Each item corresponds to an error type with yes/no answers and is assigned a different weight (from {30, 20, 20, 10, 10, 5, 5} accordingly) to form individual item scores. The total score is calculated as Total_score $= 100 - \sum_{i=1}^{7} S_i \times W_i$, where $S_i$ is error score of the $i$-th item and $W_i$ is the corresponding weight. With these defined scoring rules, GPT-4 can be prompted to score reports in accordance with these criteria, as elaborated below.

**Generating Scoring Training Dataset**  With a defined scoring system, we craft prompts that encapsulate the evaluation criteria, guiding GPT-4 to assess radiology reports similarly to human evaluators. An example of a prompt can be found in the supplementary material. Utilizing the GPT-4 API, we generate reports of varying quality based on a randomly selected subset of ground-truth reports from the MIMIC-CXR dataset. For RadCliQ scoring, we randomly select around 8000 ground-truth reports, each leading to three GPT-4-generated reports reflecting varied error levels, i.e., 0-2 errors, 3-4 errors, and 5-6 errors. Each generated report is assessed for the total number of errors as well as individual error scores. Similarly, for the MRScore scoring system, we randomly select 1800 ground-truth reports, each with three GPT-4-generated reports corresponding to three quality tiers (0-40, 40-70, and 70-100). Each report is evaluated for both total quality and individual item scores. We verified the quality of our training data by randomly selecting 50 GPT-4 generated training samples and having them evaluated by an experienced radiologist. The accuracies (accuracy = Total number of score samples that match human ratings / Total number of score samples) are 0.9 for Impression, 0.98 for Impression Organ, 0.86 for Description of Lesion, 0.92 for Clinical History, 0.98 for Completeness, 1.0 for Grammar, and 1.0 for Medical Terminology.

## 2.2 LLM-based Reward Model

ER$^2$Score is our innovative evaluation metric designed to be versatile across various evaluation frameworks. This LLM-based reward model leverages a pretrained language model, such as Llama3 (Touvron et al., 2023), fine-tuning it to match human evaluations using pairs of reports. The core of ER$^2$Score is its training process, which involves pairs of reports generated from the same ground-truth report but with different qualities. This pairing mechanism is essential for calibrating the model to distinguish between different quality levels effectively. During training, the model learns to assign higher rewards to the high-quality reports while simultaneously generating multiple individual criterion scores. These criterion scores are critical as they provide detailed insights into specific aspects of the report's quality. At the inference stage, the model predicts rewards for each individual criterion. These rewards are then summed to generate the final ER$^2$Score. To ensure precise differentiation, we introduce a scoring margin for each criterion and the overall score. This margin enables the model to recognize and learn subtle differences in report quality, enhancing its evaluative capability.

**Model Input**  Our model requires paired reports and their score margins as input. Each pair consists of an "accepted" report and a "rejected" report, both derived from the same ground-truth report, with the "accepted" report having a higher GPT-4 score than the "rejected" one. Figure 2 illustrates the pairing rule, showing the selection process for accepted and rejected reports and the calculation of their respective margins. In the example shown in Figure 2, a scoring system with four individual evaluation items is used. Accepted and rejected reports are determined based on their total scores. These reports, along with their ground-truth report, are then incorporated into a text prompt to fine-tune the LLM model for report assessment. In addition to the reports, we calculate a list of margins for both the four sub-scores and the total score: $margin^i = score^i_{accept} - score^i_{reject}$, where $i = 1, \cdots, 5$ with $i = 5$ corresponding to the total score and $i = 1, \cdots, 4$ for sub-scores. A larger margin indicates a more pronounced quality discrepancy between the two reports, while a smaller margin suggests a lesser difference. Note that although the margin of the total score is always greater than 0, the margins of the sub-scores are not necessarily positive.

**LLM Model**  Our reward model, based on the Llama3 (Meta, 2024) backbone, incorporates a multi-reward head to generate the ER$^2$Score. Llama3 was selected for its exceptional language comprehension with just 6.8M trainable parameters over 7 billion parameters in total. The multi-reward head is a linear projection layer mapping Llama-3's last layer feature map to an $N \times 1$ vector, where $N$ is the total number of sub-scores. This model is fine-tuned using Low-Rank Adaptation (LoRA) (Hu et al., 2022) for parameter-efficient fine-tuning (PEFT), allowing effective fine-tuning with minimal parameter changes. Training pairs of "accepted" and "rejected" reports calibrate the model for reward prediction. During training, the model learns to distinguish high-quality from low-quality reports by adhering to a scoring margin reflecting quality differences. Sub-scores discern quality differences per report aspect, with their summation producing the final quality assessment for generated reports.

**Objective**  Our multi-reward model aims to mimic human judgement via GPT-4 by optimizing a function based on the GPT-4 rankings of radiology reports. It discerns and predicts the preferred report within each pair, capturing subtle differences that distinguish superior reports. Instead of

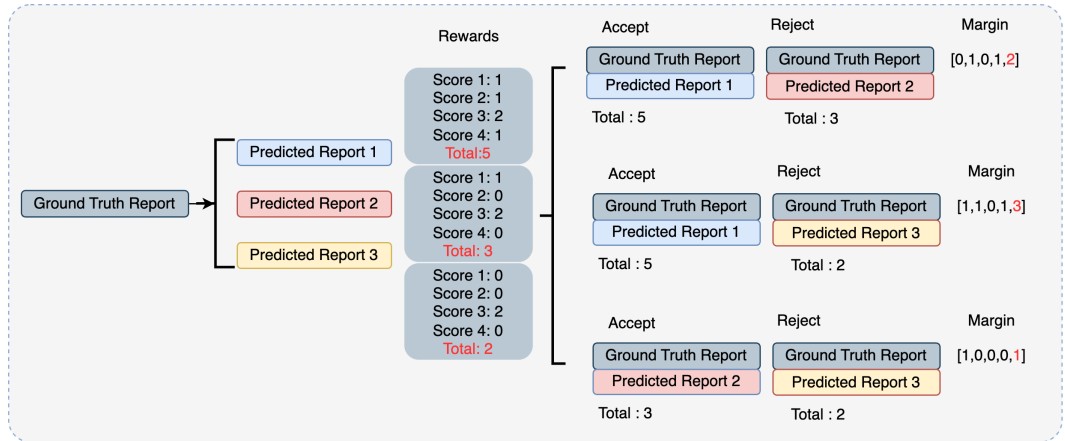

Figure 2: An illustration of report pairing rule, taking a scoring system with 4 criteria as an example.

rewarding based merely on the whole report, our objective function is devised to learn also the preference per individual criterion. The objective function is elaborated in Section 2.3. Through our objective function, we can effectively utilize the total margin to control the overall quality of the report and also respect each sub-score's margin to manage the differences in sub-scores across different overall quality levels. By adjusting the size of the margin, corresponding penalties are applied, thus training the model to produce appropriate rewards.

### 2.3 Margin Reward Enforcement(MRE) Loss Function

Considering a pair of generated reports $< y_w^i, y_l^i >$ [1] corresponding to the same $i$-th ground truth report $x^i$, the accepted report $y_w^i$ receives a higher GPT-4 score $s_w^i$ and the rejected report $y_l^i$ a lower GPT-4 score $s_l^i$. Let $s_w^{i,j}$ and $s_l^{i,j}$ denote the $j$-th sub-score of $s_w^i$ and $s_l^i$, respectively, where $j = 1, \cdots, N$ and $N$ is the number of sub-scores for a specific scoring system. Note that although the total score $s_w^i$ is greater than $s_l^i$, the sub-score $s_w^{i,j}$ is not necessarily greater than $s_l^{i,j}$. Our objective is to train the model to discern the rankings of both individual and total scores of the report pair, formulated as follows:

$$\mathcal{L}_{ind}(y_w^i, y_l^i) = \frac{1}{N} \sum_{j=1}^{N} \mathbb{1}(s_w^{i,j} \neq s_l^{i,j}) ReLU(-t_w(r_w^{i,j} - r_l^{i,j}) + t_w m^{i,j})$$

$$+ (1 - \mathbb{1}(s_w^{i,j} \neq s_l^{i,j})) ReLU(|r_w^{i,j} - r_l^{i,j}| - c),$$

$$\mathcal{L}_{tot}(y_w^i, y_l^i) = ReLU(-(\sum_{j=1}^{N} r_w^{i,j} - \sum_{j=1}^{N} r_l^{i,j}) + m^i),$$

$$\mathcal{L}_{MRE} = \sum_{i=1}^{K} \mathcal{L}_{ind}(y_w^i, y_l^i) + \lambda \mathcal{L}_{tot}(y_w^i, y_l^i). \tag{1}$$

Here $r_w^{i,j}$ and $r_l^{i,j}$ denote the $j$-th individual rewards assigned to the reports $y_w^i$ and $y_l^i$, respectively. The margin between the total scores $s_w^i$ and $s_l^i$ is denoted by $m^i = s_w^i - s_l^i$, where $m^i > 0$. The individual "margin" $m^{i,j} = s_w^{i,j} - s_l^{i,j}$ is not necessarily positive. The variable $t_w$ acts as a flag: $t_w = 1$ if $m^{i,j} > 0$, otherwise $t_w = -1$. The function $\mathbb{1}(\cdot)$ is an indicator function, returning 1 when the event occurs and 0 otherwise. $K$ is the total number of report pairs.

Our overall loss $\mathcal{L}_{overall}$ comprises two terms: the individual reward loss $\mathcal{L}_{ind}$ and the total reward loss $\mathcal{L}_{tot}$, balanced by the hyperparameter $\lambda$. An analysis of the model's behavior is as follows. For the individual reward loss $\mathcal{L}_{ind}$, if the ground truth scores have the relationship of $s_w^{i,j} > s_l^{i,j}$, i.e.,

---

[1]Here "w" stands for "win", indicating the accepted report, and "l" for "lose", indicating the rejected report.

$m^{i,j} > 0$, a penalty is incurred when the reward $r_l^{i,j}$ is larger than $r_w^{i,j} - m^{i,j}$; if $s_w^{i,j} < s_l^{i,j}$, i.e., $m^{i,j} < 0$, a penalty is incurred when the reward $r_l^{i,j}$ is smaller than $r_w^{i,j} - m^{i,j}$; if $s_w^{i,j} = s_l^{i,j}$, a penalty is incurred when the absolute difference between the two rewards is larger than a preset small positive value $c$. In addition to minimizing the individual reward loss, we also regularize the total reward loss $\mathcal{L}_{tot}$, i.e., when the total reward $\sum_j r_l^{i,j}$ of the rejected report $y_l^i$ is larger than $\sum_j r_w^{i,j} - m^i$, a penalty is incurred. Minimizing $\mathcal{L}_{overall}$ ensures that our model furnishes both individual and total scores, thereby offering nuanced insights into the assessment results.

## 3 EXPERIMENTS AND RESULT

### 3.1 DATASETS

We evaluated the effectiveness of ER$^2$Score by assessing its alignment with expert radiologist evaluations, ensuring that its predictions correlate closely with those of human experts. Our evaluation involved two datasets, ReXVal (Yu et al., 2023b) and Rad-100, each based on a distinct scoring system as described in Section 2.1. This approach allowed us to validate ER$^2$Score across different evaluative standards, exhibiting the model's adaptability to diverse assessment systems.

**ReXVal** Dataset is a publicly accessible dataset that features six board-certified radiologists' evaluations of automatically generated radiology reports. It provides a comprehensive breakdown of clinically significant and insignificant errors across six distinct categories relative to the ground-truth reports drawn from the MIMIC-CXR dataset, i.e., the RadCliQ scoring system named in our paper. The dataset encompasses 200 pairs of candidate and ground-truth reports, derived from 50 studies, each generating four candidate reports. ReXVal is primarily utilized to assess the correlation between automated metric scores and human radiologist judgments, explore the limitations of current automated metrics, and develop an integrated metric for evaluating radiological report generation.

**Rad-100** Dataset, which we developed using the MRScore scoring system, consists of 100 diagnostic reports generated by the conventional R2Gen models. Each report displays varying qualities when compared to its corresponding ground-truth report, which has been randomly sampled from the MIMIC-CXR dataset. Employing this scoring system, an experienced radiologist performs detailed evaluations of each report, assessing both overall performance and individual criteria. These evaluations provide a robust foundation for validating our ER$^2$Score.[2]

### 3.2 PERFORMANCE ON REXVAL DATASET

**Correlation Analysis of Sub-criteria** Table 1 provides a quantitative evaluation of ER$^2$Score on the ReXVal dataset, specifically constructed based on the RadCliQ Scoring System. This assessment highlights significant alignment between ER$^2$Score evaluations and expert radiologist judgments across various error categories, using Kendall's Tau and Spearman Correlation coefficients as metrics. Notably, the high correlation scores in categories such as "False prediction of a finding" (Kendall's Tau: 0.680, Spearman: 0.842) and "Omission of a finding" (Kendall's Tau: 0.507, Spearman: 0.673) demonstrate ER$^2$Score's capability in accurately identifying common radiological errors, indicating its effectiveness in recognizing significant or typical lesions. Although ER$^2$Score demonstrates strong correlations across most sub-criteria, there are areas for improvement. For example, the scores for "Incorrect location or position of a finding" (Kendall's Tau: 0.246, Spearman: 0.327) are relatively low, possibly because location and position details are often subtle and challenging to capture accurately. It is worth noting that this also highlights the advantage of ER$^2$Score over methods that provide only an overall score (Yu et al., 2023a; Zhang et al., 2019; Jain et al., 2021). By providing scores for each sub-criterion, ER$^2$Score allows us to clearly identify specific areas where the model can be enhanced.

The statistical significance of the results is underscored by extremely low p-values across all categories, reinforcing the robustness of the correlation between ER$^2$Score and expert evaluations. The overall high scores—0.751 for Kendall's Tau and 0.910 for Spearman Correlation—further validate the reliability of ER$^2$Score as an evaluation tool, highlighting its potential utility in clinical and research settings for assessing radiology reports.

---

[2]The Rad-100 dataset is entirely distinct from the datasets used for training our reward model.

Table 1: Human Correlations of $ER^2$Score on ReXVal Dataset using RadCliQ scoring criteria.

| Criteria | Kendall's Tau↑(P-Value↓) | Spearman↑(P-Value↓) |
|---|---|---|
| - False prediction of a finding | 0.680 (9.0e-41) | 0.842 (6.2e-55) |
| - Omission of a finding | 0.507 (4.9e-23) | 0.673 (8.8e-28) |
| - Incorrect location or position of a finding | 0.246 (5.9e-6) | 0.327 (2.4e-6) |
| - Incorrect severity of a finding | 0.443 (4.6e-16) | 0.569 (1.5e-18) |
| - Mention of a comparison absent in the reference impression | 0.433 (4.6e-15) | 0.545 (7.3e-17) |
| - Omission of a comparison that notes a change from a previous study | 0.267 (1.4e-6) | 0.345 (5.7e-07) |
| Total | 0.751 (4e-52) | 0.910 (5e-76) |

**Comparison with other metrics**   Table 2 compares the performance of different metrics using Kendall's Tau and Spearman correlation on ReXVal Dataset. The comparison is based on the total score. Please note that unlike $ER^2$Score, *the existing metrics have no way to be customized to user-specific sub-criteria,* making the comparison of sub-scores impossible.

We evaluate our $ER^2$Score against various Natural Language Generation (NLG) metrics, including BLEU-4 (Papineni et al., 2002), ROUGE-L (Lin, 2004), METEOR (Banerjee & Lavie, 2005), and CIDEr (Vedantam et al., 2015), as well as clinical metrics like BERTScore (Zhang et al., 2019) and RadGraph F1 (Jain et al., 2021). We also compare with RadCliQ-based metrics (Yu et al., 2023a) derived from human-annotated error scores.

The table demonstrates that $ER^2$Score exhibits a strong alignment with human judgments, as evidenced by its Kendall's Tau value of 0.751 and Spearman correlation of 0.910, both surpassing all other evaluated metrics. For instance, traditional NLG metrics like BLEU-4, ROUGE-L, and METEOR show lower correlations, with BLEU-4 achieving a Kendall's Tau of 0.345 and a Spearman correlation of 0.475. Similarly, clinical metrics such as BERTScore and RadGraph F1, while performing better than traditional NLG metrics, still fall short compared to $ER^2$Score. BERTScore, for example, has a Kendall's Tau of 0.507 and a Spearman correlation of 0.677. Notably, the RadCliQ-v1 metric shows higher correlation values, with a Kendall's Tau of 0.631 and a Spearman correlation of 0.816, indicating its effectiveness in aligning with human evaluations. However, our $ER^2$Score outperforms all these metrics, highlighting its superior ability to capture the nuances of radiology report generation as judged by experts.

Table 2: Human Correlation Comparison of Evaluation Metrics on ReXVal Dataset

| Metric | Kendall's Tau↑(P-Value↓) | Spearman↑ (P-Value↓) |
|---|---|---|
| BLEU-4 (Papineni et al., 2002) | 0.345 (2.2e-12) | 0.475 (1.2e-12) |
| ROUGE-L (Lin, 2004) | 0.491 (2.9e-23) | 0.663 (1.2e-26) |
| METEOR (Banerjee & Lavie, 2005) | 0.464 (8.4e-21) | 0.627 (2.8e-23) |
| CIDEr (Vedantam et al., 2015) | 0.499 (4.5e-24) | 0.664 (8.9e-27) |
| BertScore (Zhang et al., 2019) | 0.507 (4.5e-25) | 0.677 (3.9e-28) |
| RadGraphF1 (Jain et al., 2021) | 0.516 (4.3e-25) | 0.702 (4.4e-31) |
| semb_score (Yu et al., 2023a) | 0.494 (1.0e-23) | 0.665 (6.2e-27) |
| RadCliQ-v1 (Yu et al., 2023a) | 0.631 (6.9e-38) | 0.816 (6.6e-49) |
| $ER^2$Score (Ours) | **0.751 (4.0e-52)** | **0.910 (5.0e-76)** |

## 3.3 PERFORMANCE ON RAD-100 DATAEST

**Accuracy analysis of sub-criteria**   Since the scoring system used by Rad-100 is a binary format where the presence of an error is marked as 1 and the absence as 0 (check supplementary for detail), the results are multiplied by pre-defined weights before forming the final score. Accordingly, we evaluate the accuracy of binary classification for each sub-criterion, as reported in Table 3.

Table 3: Accuracy of Different Sub-scores in Rad-100 test dataset. Here, 'Imp. Cons.' stands for Impression Consistency, 'Imp. Org.' for Impression Organ, 'Desc. Les.' for Description of Lesion, 'Clin. Hist.' for Clinical History, 'Comp.' for Completeness, 'Gram.' for Grammar, and 'Med. Term.' for Medical Terminology.

| Sub-criteria | Imp. Cons. | Imp. Org. | Desc. Les. | Clin. Hist. | Comp. | Gram. | Med. Term. |
|---|---|---|---|---|---|---|---|
| **Accuracy** | 0.589 | 0.730 | 0.770 | 0.410 | 0.380 | 0.980 | 0.720 |

**Comparison with other metrics**    Table 4 provides a performance comparison of metrics using Kendall's Tau and Spearman correlation on the Rad-100 dataset. Similar to the previous analysis on the ReXVal dataset, we evaluate our ER²Score against various NLG and clinical metrics. As observed, on the Rad-100 dataset, our ER²Score demonstrates superior performance, with a Kendall's Tau of 0.230 and a Spearman correlation of 0.293, both statistically significant with a p-value of 0.003.

Table 4: Human Correlation Comparison of Evaluation Metrics on Rad-100 Dataset

| Metric | Kendall's Tau↑(P-Value↓) | Spearman↑(P-Value↓) |
|---|---|---|
| BLEU-4 (Papineni et al., 2002) | 0.07 (0.49) | 0.05 (0.51) |
| ROUGE-L (Lin, 2004) | 0.16 (0.10) | 0.12 (0.10) |
| METEOR (Banerjee & Lavie, 2005) | 0.11 (0.27) | 0.08 (0.26) |
| CIDEr (Vedantam et al., 2015) | 0.04 (0.70) | 0.03 (0.65) |
| BertScore (Zhang et al., 2019) | 0.13 (0.19) | 0.09 (0.20) |
| RadGraphF1 (Jain et al., 2021) | 0.09 (0.38) | 0.06 (0.43) |
| semb_score (Yu et al., 2023a) | 0.01 (0.94) | 0.01(0.94) |
| RadCliQ-v1 (Yu et al., 2023a) | 0.08(0.44) | 0.06 (0.45) |
| Ours(ER$^2$Score) | **0.23 (0.003)** | **0.29 (0.003)** |

## 3.4    PERFORMANCE COMPARISON OF LLM BACKBONES

Table 5 presents a performance comparison of various LLM backbones. Notably, Llama3 demonstrates superior performance with a medium size of trainable parameters. To ensure the scoring system is easily deployable, we focused on models with 7 billion parameters in total or fewer.

Table 5: Ablation Study of LLM Backbones on ReXVal Dataset

| Model | Trainable Params (%) | Kendall Tau (↑)) | Spearman (↑) |
|---|---|---|---|
| Llama3  (Meta, 2024) | 6.8M (0.090) | 0.751 | 0.910 |
| Vicuna-7b (Chiang et al., 2023) | 8.4M (0.127) | 0.738 | 0.901 |
| Meditron  (Chen et al., 2023) | 8.4M (0.127) | 0.709 | 0.880 |
| Gemma-7b  (Gemma Team et al., 2024) | 6.4M (0.075) | 0.707 | 0.876 |
| Qwen1.5-7b(Bai et al., 2023) | 8.4M (0.110) | 0.684 | 0.858 |
| Phi-2  (Li et al., 2023) | 5.3M (0.196) | 0.591 | 0.784 |

## 3.5    ABLATION STUDY ABOUT LOSSES AND HYPERPARAMETER

The loss we proposed comprises two terms: the individual reward loss $L_{\text{ind}}$ and the total reward loss $L_{\text{tot}}$. An ablation of the loss functions is given in Table 6. As shown, if we train $L_{\text{tot}}$ alone for predicting sub-scores, the Kendall-tau will drop from 0.751 to 0.740 for the total score, a sum of the sub-scores. If we train $L_{\text{ind}}$ alone, the Kendall-tau will drop from 0.751 to 0.738, demonstrating the effectiveness of the regularization from $L_{\text{tot}}$.

Our loss function involves two hyper-parameters: the hyperparameter $c$ is just a small positive rounding number when judging whether $r_w$ equals $r_l$, which we set to 1e-2. The hyperparameter $\lambda$ balances the two loss terms $L_{\text{ind}}$ and $L_{\text{tot}}$ and we examined its effect through the ablation study

Table 6: Spearman and Kendall correlation coefficients for different methodologies

| $\mathcal{L}_{tot}$ | $\mathcal{L}_{ind}$ | Spearman ($\uparrow$) | Kendall Tau ($\uparrow$) |
|---|---|---|---|
| $\checkmark$ | | 0.899 | 0.740 |
| | $\checkmark$ | 0.899 | 0.738 |
| $\checkmark$ | $\checkmark$ | 0.910 | 0.751 |

shown in Table7. As can be seen, our model is insensitive to $\lambda$. When it varies in a reasonably large range, our model produces better human-correlations than the existing evaluation metrics.

Table 7: Spearman and Kendall correlation coefficients with varying $\lambda$ values

| $\lambda$ | 0.5 | 0.8 | 1.0 | 1.2 | 2.0 | 3.0 |
|---|---|---|---|---|---|---|
| Spearman | 0.904 | 0.906 | 0.910 | 0.900 | 0.895 | 0.893 |
| Kendall | 0.743 | 0.746 | 0.751 | 0.740 | 0.735 | 0.729 |

### 3.6 QUALITATIVE ANALYSIS

A visual example is provided in Figure 3, demonstrating how the ER$^2$Score correlates with human ratings using the RadCliQ scoring system. As shown, the generated report inaccurately describes the severity of the "left pleural effusion" (highlighted in red), resulting in a high ER$^2$Score for "incorrect severity of a finding", which aligns with the human rating. Additionally, the report erroneously mentions a "right pleural effusion", leading to an "incorrect location/position of a finding", again perceived similarly by both the ER$^2$Score and human ratings. Lastly, the generated report fails to mention the "left retrocardiac opacification", leading to a score of '1.0' for "false prediction of a finding" from both the ER$^2$Score and the human rating.

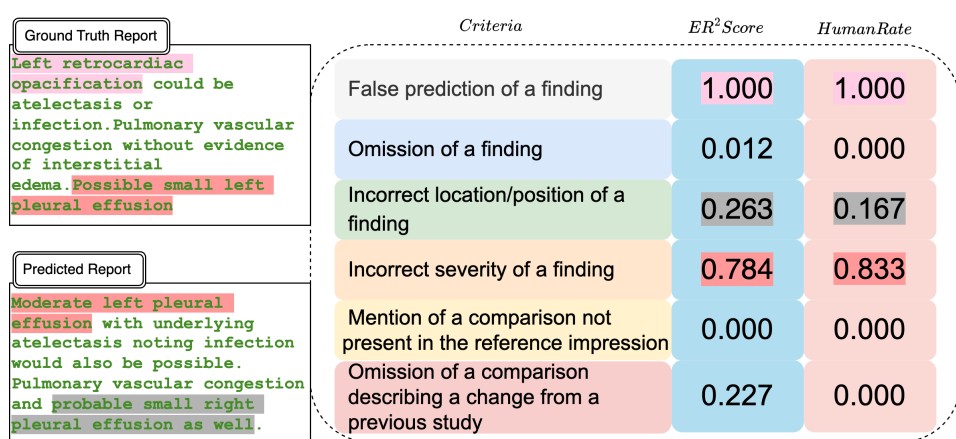

Figure 3: An visual example of ER$^2$Score from ReXVal Dataset. The highlighted sentences in reports and their corresponding scores share the same colors.

## 4 RELATED WORK

### 4.1 EVALUATION METRICS FOR RADIOLOGY REPORTS

Radiology report metrics can be categorized as language metrics and clinical metrics.

**Language Metrics** for radiology report evaluations typically rely on structured assessments and direct comparison metrics. Common approaches like BLEU (Papineni et al., 2002), ROUGE (Lin, 2004), and METEOR (Banerjee & Lavie, 2005) scores assess the textual similarity between the generated reports and a set of reference reports, focusing on aspects like n-gram overlap, precision, and recall.

Other metrics like BERTScore (Zhang et al., 2019) are calculated using embedding generated by pre-trained models to measure the similarity between the ground truth report and the generated report. However, these methods have significant drawbacks. Firstly, they often do not capture the clinical relevance or the diagnostic accuracy of the content, as they primarily focus on linguistic features rather than medical correctness. Furthermore, when applied to evaluating text generated by large language models (LLMs), such as those based on GPT architectures, these traditional metrics fall short. The complexity and variability of text generated by LLMs mean that simple lexical or syntactic comparisons are insufficient. LLMs can generate clinically plausible text that may be lexically varied but semantically similar to the reference standards. This variability can lead to evaluations that are not reflective of actual clinical usability or accuracy.

**Clinical Metrics** focus more on the clinical description in the radiology report. One prevalent metric in contemporary research is CheXpert (Irvin et al., 2019), which mandates the extraction and labeling of 14 pathological entities as 'present,' 'absent,' or 'uncertain.' The accuracy of these labels is typically assessed using tools like CheXbert, which also utilizes cosine similarity from embeddings as a metric. Another common method is RadGraph (Jain et al., 2021), which identifies clinical entities and their relationships within reports. However, these extraction-based techniques are constrained by a fixed set of entities and strict matching rules, which can lead to issues with coverage and difficulty addressing the ambiguous cases often found in reports. Although some hybrid approaches, such as RadCliQ and RadEval, attempt to amalgamate various metrics, they too fall short of fully capturing the nuances of clinical descriptions due to the inherent limitations of extraction-based methods.

## 4.2 LARGE LANGUAGE MODEL FOR EVALUATION

Previous research such as G-Eval (Liu et al., 2023) and LLM Evaluation (Chiang & Lee, 2023) has explored the use of large language models (LLMs) as automatic evaluators for language generation tasks, showing that their performance varies across different tasks. But, those are all focused on general language generation tasks. Recently, an LLM-Radjudge (Wang et al., 2024) was proposed and can use LLM to evaluate the radiology report. However, this model generally provides only a single overall score, lacking detailed interpretability. Our proposed model addresses this limitation by not only adapting to various evaluation criteria but also by breaking down scores into granular components. This enhances interpretability, allowing users to understand which specific aspects of a report contributed to its overall score. We also show our method has a high correlation with humans.

## 5 CONCLUSION

ER$^2$Score, for the first time, offers an explainable metric for evaluating radiology report. It allows for more fine-grained scoring, aligning each item of the evaluation rule with its respective sub-score, therefore enhancing the interpretability of assessment results. Leveraging GPT-4's human-like scoring capacity, we have tailored extensive training samples to fine-tune LLMs towards discerning report qualities using our designed reward loss. Our metric's adaptability allows for accommodating various scoring criteria.

Our method has the following **limitations**. First, the current level of explainability could be enhanced by incorporating detailed paragraph explanations, which are currently not included. Second, due to the costly nature of human evaluation, the scale of the test sets in this study remains limited. Third, while MIMIC-CXR is a comprehensive benchmark for chest X-rays, potential biases in the dataset could affect our model, warranting further exploration in future work.

## 6 ETHICS STATEMENT

Our ER$^2$Score model, which fine-tunes LLAMA-3 as a reward system, operates entirely locally once trained, eliminating the need for any interactions with GPT-4 during inference. This local deployment ensures that there is no risk of information leakage. GPT-4 is only used to generate training data from MIMIC-CXR dataset. MIMIC-CXR is a public dataset, which has been anonymized and de-identified. The platform Azure OpenAI is HIPAA compliant and ensures the privacy and compliance of medical data (e.g., the data are not accessible to OpenAI).

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

# A  APPENDIX / SUPPLEMENTAL MATERIAL

## A.1  EXPERIMENTS COMPUTE RESOURCES

All our models are trained by one A6000 GPU with 48 GB memory.

## A.2  LABELED DATA PROMPT DESIGN AND RESULTS

**Prompt for RadCliQ Scoring System**

Given a ground truth diagnostic report, generate three similar predicted reports. These
predicted reports should be rated based on the following error−counting rules. The reports should
   have three levels of errors:
1st level has 0 or 1 errors,
2nd level has 2, 3 or 4 errors,
3rd level has 5 or 6 errors.

Error counting rule:
Given a ground truth diagnostic report and a predicted report, score the predicted report based on
   these error categories, each with significant and non−significant errors:
− False prediction of a finding
− Omission of a finding
− Incorrect location/position of a finding
− Incorrect severity of a finding
− Mention of a comparison not present in the reference impression
− Omission of a comparison describing a change from a previous study

For each error category, assign 1 point for significant errors and 1 point for non significant errors.
   The final score is the sum of these points.

Please generate three predicted reports for the given ground truth report. After that, score  these
   three pairs (each predicted report with the ground truth) based on the error categories
   mentioned.

ground_truth_report: impression: no acute cardiopulmonary process. Cardiomegaly findings:
   frontal and lateral chest radiographs demonstrate marked cardiac enlargement unchanged
   compared to. Lungs are fairly well−aerated without focal consolidation pleural effusion or
   pneumothorax. The  visualized upper abdomen is unremarkable.

Output format:
```json
{
  "ground_truth_report": "your_ground_truth_report_here",
  "predicted_reports": [
    {
      "predicted_report": "your_predicted_report_1",
      "errors": {
        "false_prediction": "your_score",
        "omission": "your_score",
        "incorrect_location": "your_score",
        "incorrect_severity": "your_score",
        "comparison_not_present": "your_score",
        "omission_of_comparison": "your_score"
      },
      "total_score": "your_total_score_1"
    },
    {
      "predicted_report": "your_predicted_report_2",
      "errors": {
```

```
"false_prediction": "your_score",
"omission": "your_score",
"incorrect_location": "your_score",
"incorrect_severity": "your_score",
"comparison_not_present": "your_score",
"omission_of_comparison": "your_score"
},
"total_score": "your_total_score_2"
},
{
"predicted_report": "your_predicted_report_3",
"errors": {
"false_prediction": "your_score",
"omission": "your_score",
"incorrect_location": "your_score",
"incorrect_severity": "your_score",
"comparison_not_present": "your_score",
"omission_of_comparison": "your_score"
},
"total_score": "your_total_score_3"
}
]
}
```
Please directly output the json file, no other contents

**Prompt for MRscore Scoring System**

You are a skilled radiologist tasked with following task:
First Task:
By providing you with a "ground truth" report, generate three different reports,
each with a score falling within specified score ranges. The scoring rules are detailed under the second task.
The score ranges for the reports are as follows: the first report scores between 0 to 40 points,
the second report scores between 40 to 70 points, and the third report scores between 70 to 100 points.
Please generate a wider dispersion of scores

Second Task:
Evaluate radiology reports,
The three generated reports mentioned above are referred to as "predicted reports," and each is paired with the given "ground truth" report. Therefore, we will evaluate the three pairs of reports based on the following rules.
To achieve this objective, we compare the predicted report with the ground truth report to identify discrepancies between them. These discrepancies are defined according to the 'Error category' described in the table below, with each error assigned a specific weight. Upon the detection of an error, the weight is deducted from the total score of 100 according to the corresponding rule as follows. Analysis why
The scoring rule is:
− Check the predicted report for the presence of error items listed in the table below. Each item in the table needs to be checked, and if an error item is found, locate the corresponding score for this error item in the table and note it down, subtracting it from 100.
− Based on all the errors found, calculate all the error scores to get the total score, which means subtracting all existing error scores from 100.
− For the first error item under 'impression consistency', if there is no impression section in the ground truth, then this item does not count towards the score. Skip it with no score subtraction and proceed to analyze the other items in the table.
− Please generate the score and the analysis separately

Please format the result in a JSON format

| Error Category | Description | Score |
|---|---|---|
| Impression consistency | The impression shows normal or abnormal | 30 |
| Impression Organ | Is Lesion related Anatomical organ correct | 20 |
| Description of Lesion | Check the correctness of lesion location, lesion size, lesion opacity, Cardiovascular size, bone integrity | 20 |
| Clinical History | Check the correctness of Operation history, treatment, family history | 10 |
| Completeness | Conclude all information in ground truth report | 10 |
| Grammar | Vocabulary spelling, fluently | 5 |
| Medical Terminology | Non−medical related terminology | 5 |

Final Score = 100 − sum(Error Weight)
The given ground truth report is:
{content}

Output format example is as follows, if there is an error in the above rule, mark the corresponding score in the scoring part of the JSON
Predicted Report 0−40 indicates the quality of the generated report falls within the score bracket [0,40] For example 35
Predicted Report 40−70 indicates the quality of the generated report falls within the score bracket [40,70] For example 60
Predicted Report 70−100 indicates the quality of the generated report falls within the score bracket [70,100] For example 85

```
{{
  "Ground Truth Report": {content},
  "Predicted Report 0−40": {{
    "Ground Truth Report": "",
    "Predicted Report": "",
    "Scoring": {{}},
    "Analysis": {{}},
    "Final Score":
  }},

  "Predicted Report 40−70": {{
    "Ground Truth Report": "",
    "Predicted Report": "",
    "Scoring": {{}},
    "Analysis": {{}},
    "Final Score":
  }},

  "Predicted Report 70−100": {{
    "Ground Truth Report": "",
    "Predicted Report": "",
    "Scoring": {{}},
    "Analysis":{{}},
    "Final Score":
  }}
}}
```

A.3 LABELED DATA

**Labeled data for RadCliQ Scoring System** In our study, we using LLM generated the labeled data as follows:

```
{
  "ground_truth_report": "Impression: Bibasilar atelectasis. Tortuous aorta with likely ascending
```

aortic aneurysm, unchanged, which can better be assessed with contrast−enhanced CT.
    Emphysema.
Findings: Heart size is normal. Markedly tortuous thoracic aorta is again noted with dilatation
of the ascending aorta concerning for aneurysm, overall unchanged. The pulmonary vascularity
is not engorged. Calcified granulomas are noted bilaterally, the largest of which is in the
left upper lobe, unchanged. Streaky linear opacities are again noted within the lung bases
compatible with atelectasis. Lungs are hyperinflated with emphysematous changes again noted.
No focal consolidation, pleural effusion or pneumothorax is present. No acute osseous
abnormality is seen.",
"predicted_reports": [
  {
    "predicted_report": "Impression: Mild bibasilar atelectasis. Aorta is tortuous with a
    possible ascending aortic aneurysm, showing progression, which should be evaluated with
    contrast−enhanced CT. Mild emphysema noted. Findings: Heart size appears slightly enlarged.
    The thoracic aorta shows tortuosity without clear evidence of aneurysm. No significant
    change in pulmonary vascularity. Multiple calcified granulomas are observed, with a notable
    one in the right lower lobe. Linear opacities suggest atelectasis at the lung bases. Lungs
    show signs of hyperinflation, indicative of emphysema. There is no evidence of consolidation,
    pleural effusion, or pneumothorax. No bone abnormalities detected.",
    "errors": {
      "false_prediction": "2",
      "omission": "1",
      "incorrect_location": "1",
      "incorrect_severity": "2",
      "comparison_not_present": "0",
      "omission_of_comparison": "1"
    },
    "total_score": "7"
  }
]
}

**Labeled data for MRscore Scoring System**

{
  "Ground Truth Report": "Lateral view somewhat limited due to overlying motion artifact. The
    lungs are low in volume.  There is no focal airspace consolidation to suggest pneumonia.  A
    1.2−cm calcified granuloma just below the medial aspect of the right hemidiaphragm is
    unchanged from prior study.  No pleural effusions or pulmonary edema. There is no
    pneumothorax. The inferior sternotomy wire is fractured but unchanged. Surgical clips and
    vascular markers in the thorax are related to prior CABG surgery.",
  "Predicted Report 0−40": {
    "Ground Truth Report": "Lateral view somewhat limited due to overlying motion artifact. The
     lungs are low in volume.  There is no focal airspace consolidation to suggest pneumonia.  A
    1.2−cm calcified granuloma just below the medial aspect of the right hemidiaphragm is
    unchanged from prior study.  No pleural effusions or pulmonary edema. There is no
    pneumothorax. The inferior sternotomy wire is fractured but unchanged. Surgical clips and
    vascular markers in the thorax are related to prior CABG surgery.",
    "Predicted Report": "The patient's lungs are normal in volume with no signs of pneumonia.
    There is a possibility of a small pneumothorax. The sternotomy wire is intact and there are no
    surgical clips visible.",
    "Scoring": {
      "Impression consistency": 30,
      "Impression Organ": 20,
      "Description of Lesion": 20,
      "Clinical History": 10,
      "Completeness": 10,
      "Grammar": 0,
      "Medical Terminology": 0

```
        },
        "Analysis": {
          "Impression consistency": "The predicted report contradicts the ground truth report in terms
         of lung volume and presence of pneumothorax.",
          "Impression Organ": "The predicted report fails to mention the right hemidiaphragm and
         the thorax, which are present in the ground truth report.",
          "Description of Lesion": "The predicted report does not mention the 1.2−cm calcified
         granuloma and the fractured sternotomy wire.",
          "Clinical History": "The predicted report does not mention the prior CABG surgery.",
          "Completeness": "The predicted report does not include all information from the ground
         truth report.",
          "Grammar": "No issues with grammar or spelling.",
          "Medical Terminology": "No issues with medical terminology."
        },
        "Final Score": 10
      },
      "Predicted Report 40−70": {
        "Ground Truth Report": "Lateral view somewhat limited due to overlying motion artifact. The
         lungs are low in volume.  There is no focal airspace consolidation to suggest pneumonia.  A
         1.2−cm calcified granuloma just below the medial aspect of the right hemidiaphragm is
         unchanged from prior study.  No pleural effusions or pulmonary edema. There is no
         pneumothorax. The inferior sternotomy wire is fractured but unchanged. Surgical clips and
         vascular markers in the thorax are related to prior CABG surgery.",
        "Predicted Report": "Lungs are low in volume. There is no pneumonia present but there is a
         1.2−cm calcified granuloma below the right hemidiaphragm. There are no signs of pleural
         effusions or pulmonary edema. A sternotomy wire is visible but its status is unknown. There
         are surgical clips present in the thorax.",
        "Scoring": {
          "Impression consistency": 30,
          "Impression Organ": 0,
          "Description of Lesion": 20,
          "Clinical History": 0,
          "Completeness": 10,
          "Grammar": 0,
          "Medical Terminology": 0
        },
        "Analysis": {
          "Impression consistency": "The predicted report does not mention the limited lateral view
         due to overlying motion artifact.",
          "Impression Organ": "The predicted report correctly identifies the organs mentioned in the
         ground truth report.",
          "Description of Lesion": "The predicted report does not mention the fractured sternotomy
         wire.",
          "Clinical History": "The predicted report correctly mentions the surgical clips in the thorax,
         indicating a history of surgery.",
          "Completeness": "The predicted report fails to include the status of the sternotomy wire and
         the absence of pneumothorax.",
          "Grammar": "No issues with grammar or spelling.",
          "Medical Terminology": "No issues with medical terminology."
        },
        "Final Score": 40
      },
      "Predicted Report 70−100": {
        "Ground Truth Report": "Lateral view somewhat limited due to overlying motion artifact. The
         lungs are low in volume.  There is no focal airspace consolidation to suggest pneumonia.  A
         1.2−cm calcified granuloma just below the medial aspect of the right hemidiaphragm is
         unchanged from prior study.  No pleural effusions or pulmonary edema. There is no
         pneumothorax. The inferior sternotomy wire is fractured but unchanged. Surgical clips and
         vascular markers in the thorax are related to prior CABG surgery.",
```

"Predicted Report": "Limited lateral view due to overlying motion artifact. Lungs are low in volume with no signs of pneumonia. A 1.2−cm calcified granuloma is present below the right hemidiaphragm. No pleural effusions or pulmonary edema. No pneumothorax. The sternotomy wire is fractured but unchanged. Surgical clips and vascular markers indicate a history of CABG surgery.",
"Scoring": {
    "Impression consistency": 0,
    "Impression Organ": 0,
    "Description of Lesion": 0,
    "Clinical History": 0,
    "Completeness": 0,
    "Grammar": 0,
    "Medical Terminology": 0
},
"Analysis": {
    "Impression consistency": "The predicted report is consistent with the ground truth report.",
    "Impression Organ": "The predicted report correctly identifies the organs mentioned in the ground truth report.",
    "Description of Lesion": "The predicted report correctly describes the lesions mentioned in the ground truth report.",
    "Clinical History": "The predicted report correctly identifies the patient's clinical history.",
    "Completeness": "The predicted report includes all information from the ground truth report.",
    "Grammar": "No issues with grammar or spelling.",
    "Medical Terminology": "No issues with medical terminology."
},
"Final Score": 100
    }
}

## A.4 Scoring Dataset Prompt Samples

**Scoring Dataset Prompt Samples for RadCliQ Scoring System**

{
  {
      "chosen": "Human: 'The ground truth report is: Impression: Tortuous aorta with prominence of ascending aortic contour. If clinical concern, could be further evaluated with chest CT. Multiple calcified granulomas. Findings: Chest PA and lateral radiograph demonstrates a tortuous aorta with questionable prominence of the ascending aortic contour. Heart size is normal. The previously noted right lower lung opacity has largely resolved with minimal residual linear opacities evident on the lateral view, likely post−inflammatory. There has been interval resolution of the previously identified right lower lung opacity. Multiple calcified nodules identified, the largest located in the left upper lung. No pleural effusion or pneumothorax evident., Assistant: 'The predicted report is: Impression: Aorta is tortuous with an unclear prominence of the ascending contour. Chest CT is advised for further assessment. Calcified granulomas are observed. Findings: The chest X−ray reveals a tortuous aorta, but the prominence of the ascending aortic contour is not evident. The heart appears enlarged, which is a new finding. The right lower lung opacity noted before has resolved, with only minor residual opacities likely due to post−inflammatory changes. Several calcified nodules are seen, with a significant one in the right upper lung. No pleural effusion or pneumothorax is present.', Human: 'Please evaluate the quality of the assistant's predicted radiology report based on the ground truth provided.'",
      "rejected": "Human: 'The ground truth report is: Impression: Tortuous aorta with prominence of ascending aortic contour. If clinical concern, could be further evaluated with chest CT. Multiple calcified granulomas. Findings: Chest PA and lateral radiograph demonstrates a tortuous aorta with questionable prominence of the ascending aortic contour. Heart size is normal. The previously noted right lower lung opacity has largely resolved with minimal residual linear opacities evident on the lateral view, likely post−inflammatory. There

has been interval resolution of the previously identified right lower lung opacity. Multiple calcified nodules identified, the largest located in the left upper lung. No pleural effusion or pneumothorax evident., Assistant: 'The predicted report is: Impression: Aorta shows tortuosity with no significant prominence of the ascending aortic contour. Suggest chest CT if there are clinical concerns. Several calcified granulomas identified. Findings: Chest radiograph indicates a tortuous aorta without clear prominence of the ascending aortic contour. Heart size within normal limits. Previous right lower lung opacity has mostly resolved, with some residual linear opacities visible on the lateral view, suggesting post−inflammatory changes. Multiple calcified nodules seen, predominantly in the left upper lung. No signs of pleural effusion or pneumothorax detected.', Human: 'Please evaluate the quality of the assistant's predicted radiology report based on the ground truth provided.'",
    "margin": [
      1,
      0,
      1,
      0,
      0,
      0,
      2
    ],
    "chosen_score": 4,
    "rejected_score": 2
  }
},

**Scoring Dataset Prompt Samples for MRscore Scoring System**

{
  "chosen": "Human: 'The ground truth report is: Single frontal radiograph of the chest was performed and reveals no acute cardiopulmonary process. The cardiomediastinal and pleural structures are unremarkable. There is scarring in the upper lungs with superior traction of the hila. There is no pleural effusion or pneumothorax. Heart size is normal. Surgical hardware is seen at the right glenohumeral joint and ___ are seen within the abdomen with cardiophrenic angle may represent a small left pleural effusion as was previously seen approximately one month prior., Assistant: 'The predicted report is:Frontal chest radiograph shows no acute cardiopulmonary process. There is scarring in the upper lungs. No pleural effusion or pneumothorax. Heart size is normal.', Human: 'Please evaluate the quality of the assistant's predicted radiology report based on the ground truth provided.'",
  "rejected": "Human: 'The ground truth report is:Single frontal radiograph of the chest was performed and reveals no acute cardiopulmonary process. The cardiomediastinal and pleural structures are unremarkable. There is scarring in the upper lungs with superior traction of the hila. There is no pleural effusion or pneumothorax. Heart size is normal. Surgical hardware is seen at the right glenohumeral joint and ___ are seen within the abdomen with cardiophrenic angle may represent a small left pleural effusion as was previously seen approximately one month prior., Assistant: 'The predicted report is:Frontal chest radiograph shows the heart and lungs are normal. No previous surgical hardware or abnormality is noted.', Human: 'Please evaluate the quality of the assistant's predicted radiology report based on the ground truth provided.'",
  "margin": [
    0,
    20,
    0,
    0,
    0,
    0,
    0,
    20
  ],
  "chosen_score": 40,

```
    "rejected_score": 20
}
```

