# OpenReview forum: "ER2Score: An Explainable and Customizable Metric for Assessing Radiology Reports with LLM-based Rewards"
_ICLR.cc/2025/Conference — ICLR 2025 Conference Withdrawn Submission_

### Official Review · Reviewer_t2o2 · 2024-11-01

**Soundness:** 3
**Presentation:** 2
**Contribution:** 2
**Rating:** 6
**Confidence:** 4

**Summary:**

This paper introduces a method for training language models to evaluate generated radiology reports across a variety of metrics. The authors create a dataset of generated radiology reports of varying quality and train a reward model to assign scores to radiology reports using this “ground-truth” data, producing fine-grained evaluations across a range of relevant sub-categories. The authors find that their approach produces results that align better with radiologists than other common approaches. As new approaches to automated radiology report generation are developed, this approach could be an effective method of generating quick feedback on the effectiveness of these approaches.

**Strengths:**

The approach is well motivated taking an approach to evaluation that can score reports across a range of clinically valid dimensions. The described loss function seems mathematically sound with different terms well justified and the usage of human evaluation when scoring the method is important in validating the approach beyond automated metrics.

**Weaknesses:**

Despite the method outperforming other approaches, the final performance of ER2Score still seems fairly low. The manuscript needs more justification in light of this low performance to justify why this method will still be useful. It would also be useful to study the ReXVal criteria where performance was particularly low in more detail, to understand why agreement with radiologists might have been poor.

Additionally, there were areas where the approach was unclear and requires more clarification. Please see the questions.

**Questions:**

* The sample-size of 50 used to validate/spot-check the model generated radiology reports is fairly low. Did the authors perform any kind of power analysis to verify that this is a sufficiently large sample to validate their automated report generation process?
* I’m not sure I fully understand the pairing rule. From figure 2 it appears that each report in a predicted report pair was also paired with the ground-truth report? Is this the case or did a pair simply consist of 2 predicted reports? If the former then what was the reason for also providing the ground-truth report?
* For the Rad-100 dataset was there a 1-to-1 correspondence between a report in the dataset and ground-truth MIMIC report it was based on? Or are there several reports based on a single report?
* In Table 1 what does Total mean? These values seem very different to the individual criteria score
* How are our NLG metrics computed for table 2? Are these based on BLEU/ROUGE between the generated report and a ground-truth report?

---

> ### Author Response · Authors · 2024-11-14
>
> Thanks for your feedback, to address your concerns, we have outlined the following points:
>
> We have to point out our approach represents a significant improvement in providing human-consistent evaluation metric for R2Gen task: Our Kendall Tau's correlation improves from 0.63 (the second best) to 0.75, and our Spearman correlation improves from 0.81 (the second best) to 0.91 on ReXVal dataset.  Please note that, to advance R2Gen research, it is essential to develop evaluation metrics that are highly correlated with human assessments. Currently, many methods still focus on improving existing non-trainable metrics, but these metrics often have weak correlation with human evaluations. Relying on them as a guide for improvement may misdirect R2Gen research. Therefore, constructing metrics closely aligned with human judgment not only enables more accurate quality assessment of generated reports but also provides more effective guidance for research and model development in this field. Meanwhile, this is a very challenging task. We cannot hope to get a 100% correlated scoring system as humans in a single shot, but this research has to start from somewhere and we have demonstrated promising solutions.
>
> Questions:
>
> 1. Please note that the dataset of 50 samples is only used for the sanity-check of the training data; the validation of our model is conducted on two datasets with distinct criteria, comparable (indeed more than) to the scale of the existing works in this domain.
>
> 2. Each predicted report and its corresponding ground truth report must be combined as input for the evaluation process, as all evaluation metrics require both. The ground truth report serves as the reference for assessing the accuracy and quality of the predicted report.
>
> 3. The Rad-100 dataset has unique 100 ground truth reports.
>
> 4. The "Total" here represents the correlation of the report's overall score between the metrics and the human-rated total score. In contrast, the "Individual" scores indicate the correlation between each specific criterion’s score from the model and the corresponding criterion score given by humans. While the individual scores themselves are summed to produce the total score, the correlations (Total and Individual) do not have an additive relationship—only the scores do.
>
> 5. Yes, for table 2, all other NLG metrics need both generated report and ground truth report to calculate the score

---

### Official Review · Reviewer_JaGg · 2024-11-02

**Soundness:** 2
**Presentation:** 1
**Contribution:** 2
**Rating:** 3
**Confidence:** 3

**Summary:**

This paper introduces ER2Score, a novel metric for evaluating automated radiology report generation. ER2Score leverages LLMs, specifically GPT-4 and Llama 3, to address the limitations of existing metrics that rely on n-gram overlap or predefined clinical entities.

GPT-4 generates training data by scoring synthetically generated reports against reference reports based on predefined customizable criteria (RadCliQ and MRScore). Llama 3 is then LoRA fine-tuned as a reward model to predict sub-scores for each criterion, which are summed to produce the final ER2Score. The authors claimed ER2Score's improved correlation with human judgments and superior model selection capabilities compared to traditional metrics on two datasets.

**Strengths:**

The generation of sub-scores for individual evaluation criteria offers valuable insights into report quality, potentially better interpretability, and potentially enabling targeted improvements.

The ability to train the metric with different scoring systems (RadCliQ and MRScore) demonstrates flexibility and adaptability to diverse evaluation needs.

**Weaknesses:**

The relatively small size of the evaluation datasets (ReXVal and Rad-100) raises concerns about the generalizability of the results. Larger-scale evaluations are needed to validate the robustness of ER2Score.

Relying on GPT-4 generating training data, certain combinations of sub-scores might be rare, leading to data sparsity issues. This can hinder the model's ability to learn effectively and generalize to unseen examples, affecting convergence. Also, introduce a performance cap.

The supplementary material provides example prompts, but more details on the prompt engineering process and its impact on GPT-4's scoring would be beneficial. How sensitive is the performance to prompt variations?

Lacking inter-human agreement.

**Questions:**

How sensitive is ER2Score to the quality of the ground truth reports used for training and evaluation?

What are the computational costs associated with training and using ER2Score? How does it compare to other metrics?

What are the limitations of using reward modeling for this task? Is the reward guaranteed to converge due to the multi-dimensionality of the grading criteria?

Other:

line 77, citation (Meta, 2024) missing space afterward.

line 142, 143 "Kendall's tau of 0.735" - needs to be consistent capitalization: "Kendall's Tau."

Really inconsistent capitalization on Dataset: for example, line 288 "ReXVal Dataset" but "MIMIC dataset" on following lines.

Various instances: Inconsistent capitalization of "RadCliQ" and "MRScore."

And many awkward phrasings. Careful proofreading and editing are required going further.

---

> ### Author Response · Authors · 2024-11-14
>
> Thank you for the feedback
>
> 1.
> There are no such datasets publicly available. We would like to emphasize the scale of the dataset involved in our work is at the same scale of relevant works in the literature.
>
> 2.
> Different from existing methods directly predicting the overall score only, our method predicts the subscores and adds them to calculate the overall score. This indeed improves the model generalization capacity, reducing the risk of overfitting to data, because it is a more challenging task to get both the subscores and the overall score to be highly correlated to human ratings. Meanwhile, using our current training dataset, our model has significantly outperformed existing evaluation metrics in correlating to human ratings.
>
> 3.
> Please note that we used zero-shot prompting to generate the training samples. It is reasonable to expect that incorporating few-shot prompts could further enhance our model's performance, as generally agreed in this field.
>
> 4. Please note that the test dataset ReXVal involves 6 human raters and the ground-truth scores are based on inter-human agreement.
>
> Questions.
> 1. Please note that the ground-truth reports are taken from MIMIC-CXR, while the quality of the reports is controlled by the dataset. We use the same evaluation datasets as existing comparable methods in this domain.
>
> 2. About computational cost: The conventional NLG metrics and the existing clinic-scores do not need training, however, they correlate poorly with human ratings and they have no ways to customize to different evaluation criteria. These are the challenges that this work wants to address.  Compared with other trainable evaluation metrics like GREEN, our model is computationally very efficient: GREEN’s inference time is 1.06 seconds per sample, while ours is only 0.04 seconds per sample. GREEN has a much higher training cost: it was trained on 8x NVIDIA A100 Tensor Core GPUs with 40GB VRAM each, using a batch size of 2,048 for 12 epochs. In contrast, our model was trained on a single A6000 GPU with 40GB VRAM, using a batch size of 6 for 4 epochs.
>
> 3. Our training loss curve converges on the two tasks.

---

### Official Review · Reviewer_c4xF · 2024-11-02

**Soundness:** 2
**Presentation:** 2
**Contribution:** 2
**Rating:** 3
**Confidence:** 4

**Summary:**

This paper addresses the limitations of traditional metrics for evaluating automated RRG and introduces ER2Score, a metric based on LLMs. ER2Score attempts to improve evaluation accuracy by using a reward model that can be tailored to user-defined criteria and provides detailed sub-scores for enhanced interpretability.

**Strengths:**

Many experiments and ablation studies. Nicely presented paper with clear figures and methodology. I commend the authors for doing many experiments and evaluation studies to validate their metric.

**Weaknesses:**

It seems like the candidate metrics selected for comparison are not the most recent ones or relevant ones, especially given ER2Score's LLM-based nature. It is expected for ER2Score to outperform the traditional lexical/semantic/factuality metrics; however, I cannot judge on its performance without comparisons to other more recent LLM-based metrics for RRG.

I recommend the authors do experiments with GREEN (https://arxiv.org/html/2405.03595v1), FineRadScore (https://arxiv.org/html/2405.20613v2), and G-Rad (https://arxiv.org/html/2403.08002v2), as these are more relevant. They also use similar ReXVal dataset and error counts, as well as correlation evaluation with Kendall's Tau.

**Questions:**

Recommend authors do further comparison and validation studies with the suggested metrics to fully demonstrate/prove that ER2Score is superior.

---

> ### Author Response · Authors · 2024-11-14
>
> Thanks for your feedback
> Please note that, among your suggested methods, FineRadScore and G-Rad are not formally published methods and they are not directly comparable to our method, as they rely on online LLMs, which raises privacy issues and is inviable for practical use. In contrast, our model is an offline model.
>
> Compared with GREEN, our method has multiple advantages. First, our Kendall’s tau score is higher than GREEN’s (Ours: 0.751, GREEN: 0.640). GREEN adds interpretability through free-text analysis but sacrifices scoring accuracy due to the task complexity. Furthermore, GREEN’s fine-tuning of LLMs lacks a dedicated loss function as ours, limiting its sensitivity to nuanced quality differences. Second,  our inference speed is significantly faster. GREEN’s inference time is 1.06 seconds per sample, while ours is only 0.04 seconds per sample. Third, GREEN has a much higher training cost: it was trained on 8x NVIDIA A100 Tensor Core GPUs with 40GB VRAM each, using a batch size of 2,048 for 12 epochs. In contrast, our model was trained on a single A6000 GPU with 40GB VRAM, using a batch size of 6 for 4 epochs.
>
> As mentioned above, FineRadScore is not comparable to our model, as it only leverages the capabilities of large models (GPT4 and Claude-3 ) without fine-tuning, and scores each report by comparing sentences individually. Notably, it cannot provide an offline model to cater for the privacy concerns for evaluation.
>
> As mentioned above, similar to FineRadScore, G-Rad relies solely on an online GPT model as its evaluator, which is both costly and requires online access.

---

### Official Review · Reviewer_8qTi · 2024-11-04

**Soundness:** 2
**Presentation:** 2
**Contribution:** 2
**Rating:** 5
**Confidence:** 3

**Summary:**

The paper introduces ER2Score to automated radiology report generation evaluation.
ER2Score utilizes a reward model based on GPT-4, for more nuanced and human-aligned evaluations, and provides sub-scores across various criteria. The authors also train a reward model using ER2score.

**Strengths:**

- ER2Score provides detailed sub-scores for specific report components, making it easier to identify deficiencies in generated reports.
- The metric is highly agreeable with human evaluation standards

**Weaknesses:**

- Testing is restricted to a narrow set of benchmarks on a total of 200 + 100 reports, which is quite small and introduces concerns of overfitting.
- ER2Score bases its scoring criteria on established scoring systems like RadCliQ and MRScore, which are limited in scope and may not fully encompass the complexity of radiological reporting. The model lacks flexibility to adapt to other or emergent scoring needs without significant retraining.
- ER2Score's accuracy varies across sub-criteria, performing well on some (e.g., omission of findings) but poorly on others (e.g., location or position accuracy). More in-depth analysis would be interesting to see why this is the case.
- Margin-based scoring criteria for distinguishing between accepted and rejected reports is interesting, but lacks an in-depth analysis of the impact of margin size on model sensitivity. Margin thresholds should be rigorously analyzed for borderline cases.

**Questions:**

See weaknesses

---

> ### Author Response · Authors · 2024-11-14
>
> Thank you for the feedback,
> 1.
> First, we’d like to emphasize that the dataset scale in our work is comparable to that of comparable studies in the literature. Evaluating the scoring system's correlation with human ratings requires ground-truth scores from radiologists, which naturally limits the size of the test datasets. Nevertheless, we have evaluated two datasets, including the public ReXVal dataset, using two different scoring criteria, a notable advancement over existing works that rely solely on ReXVal and test only one criterion (e.g., RadCLiQ or MRScore). Could you please provide any other examples that cover more criteria or datasets than ours?
>
> \
> 2. Please note that RadCLiQ and MRScore are currently the only available scoring systems explicitly provided in the literature of relevant works. Also note that, no existing methods have evaluated both scoring systems as we did. If additional systems were available, we would apply our method to them as well. By using two distinct scoring systems, we show that our model can be easily customized to various evaluation criteria.
>
> While our model requires retraining to adapt to new scoring systems, this represents a big leap compared to existing NLG metrics and clinical scores, which have NO WAY to adapt to different evaluation standards.
>
>
> 3.
> In Table 1, although the correlation values vary across different sub-criteria, they are all statistically significant, and the overall score, which is the addition of all sub-scores, shows very high correlation. The variation of correlation values may be related to the distribution of training samples according to the sub-criteria. For example, in MIMIC-CXR, there are only a small amount of reports that have clinic history records, which may contribute to the relatively low correlation of the “omission of comparison that notes a change from a previous study.
>
> 4.
> Please note that the margin is NOT a hyperparameter, but was calculated using the difference in the scores of the training pairs (i.e., accepted and rejected reports) generated by GPT-4 . If we do not use margin (i.e., set margin = 0), the Kendall-tau correlation will drop from 0.751 to 0.731 and the Spearman correlation will drop from 0.910 to 0.895.

---

### Official Review · Reviewer_GdmL · 2024-11-05

**Soundness:** 3
**Presentation:** 3
**Contribution:** 3
**Rating:** 3
**Confidence:** 4

**Summary:**

This study proposes ER2Score, an evaluation framework for automated radiology report generation, grounded in the RadCliQ and MRScore scoring systems. To address the limitations of traditional metrics, ER2Score leverages pairs of “accepted” and “rejected” reports with corresponding scores obtained from GPT-4. These pairs are then used to train a Llama3 model, allowing it to assign quality scores to individual reports during inference. ER2Score demonstrates the potential to capture the semantic content and clinical significance of radiology reports, achieving a high correlation score with human evaluations. The authors suggest that this method offers a more fine-grained assessment compared to existing evaluation approaches.

However, while ER2Score builds on RadCliQ and MRScore, the improvements remain incremental, potentially limiting its overall contribution.

**Strengths:**

1. Development of a Model with Strong Human Alignment: ER2Score demonstrates high alignment with human evaluations by employing GPT-4 to generate high- and low-quality report samples for a large set of ground truth reports, with GPT-4 assigning scores based on the RadCliQ and MRScore evaluation criteria. These scored pairs are then used as input for training Llama3, introducing a structured approach to calculating both total and individual scores, resulting in a model that closely aligns with human assessments.

2. Rad-100 Dataset for Comprehensive Evaluation: The Rad-100 dataset, comprising 100 reports scored by an experienced radiologist following the MRScore framework, was created to provide additional validation of ER2Score’s effectiveness.

**Weaknesses:**

1. This study builds on RadCliQ and MRScore, yet the improvements are minimal, with only slight modifications to the loss function, making it difficult to view ER2Score as a truly fine-grained evaluation metric. Additionally, the customization feature highlighted as a key contribution lacks sufficient experimental support, with limited results in Table 2 that do not fully demonstrate its claimed flexibility.

2. The study would benefit from evaluating state-of-the-art models using ER2Score, providing further evidence of its practical advantages beyond metric-to-metric comparisons.

**Questions:**

N/A

---

> ### Author Response · Authors · 2024-11-14
>
> Thank you for your feedback. To address your concerns, we have outlined the following points:
>
> 1. Our model is significantly different from RadCliQ and MRsore.
>
> First, neither method can conduct fine-grained evaluation, e.g.,  by combining sub-criteria, we can clearly identify the reasons for a report’s poor quality, e.g., whether due to incorrect lesion location, incorrect severity of findings, or omission of findings, which is not achievable by either RadCliQ or MRScore.  Notably, simultaneously predicting subscores accurately is a more challenging task requiring a carefully designed optimization process. In contrast, RadCliQ combines non-trainable scores linearly, offering limited improvement and flexibility, while MRScore can only predict an overall score. Even for the overall score, we improved the Kendall's Tau from 0.63 (the second best) to 0.75 and Spearsman correlation from 0.81 (the second best) to 0.91 on the public ReXVal dataset. That is a significant improvement in any means.
>
> Secondly, we design completely different loss functions from existing LLM-based scoring systems, beyond enabling subscore evaluation.
>
> Thirdly, it is the first time that an LLM-based R2Gen scoring system can demonstrate its adaptability across different evaluation criteria.
>
> Moreover, as for "the customization feature highlighted as a key contribution lacks sufficient experimental support, with limited results in Table 2 that do not fully demonstrate its claimed flexibility", please note that Table 2 alone is not used to demonstrate the flexibility of customization feature!!! The customization feature is demonstrated by our experiments on two datasets with different scoring systems (results presented in Table 2 and Table 4).
>
> 2. To score existing R2Gen models, we need to access the reports generated by these methods, which are absent in most cases. Also, although this is an interesting work, it totally goes beyond the scope of this paper, which focuses on designing a new evaluation method and comprehensively evaluating its performance.

---

### Author Response · Authors · 2024-11-15
**A general comment**

Among my numerous paper submission experiences, this is the first time that I received scores of 1 and 2 for the presentation, possibly due to we are working on an unconventional research topic: developing human-consistent evaluation metrics for automated medical report generation methods. Hopefully, ICLR can better match reviewers next time.

---

### Note · Authors · 2024-11-15

**Comment:**

We disagree with the reviewers' comments and have submitted responses addressing our perspective. At this time, we have decided to formally withdraw our paper from consideration.

**Withdrawal Confirmation:**

I have read and agree with the venue's withdrawal policy on behalf of myself and my co-authors.